# Genomic structural variation-mediated allelic suppression causes hybrid male sterility in rice

Rongxin Shen[1,2,3], Lan Wang[2,3], Xupeng Liu[1,2,3], Jiang Wu[1,2,3], Weiwei Jin[4], Xiucai Zhao[1,2,3], Xianrong Xie[1,2,3], Qinlong Zhu[1,2,3], Huiwu Tang[1,2,3], Qing Li[1,2,3], Letian Chen[1,2,3] & Yao-Guang Liu[1,2,3]

Hybrids between divergent populations commonly show hybrid sterility; this reproductive barrier hinders hybrid breeding of the *japonica* and *indica* rice (*Oryza sativa* L.) subspecies. Here we show that structural changes and copy number variation at the *Sc* locus confer *japonica*–*indica* hybrid male sterility. The *japonica* allele, *Sc-j*, contains a pollen-essential gene encoding a DUF1618-domain protein; the *indica* allele, *Sc-i*, contains two or three tandem-duplicated ~28-kb segments, each carrying an *Sc-j*-homolog with a distinct promoter. In *Sc-j/Sc-i* hybrids, the high-expression of *Sc-i* in sporophytic cells causes suppression of *Sc-j* expression in pollen and selective abortion of *Sc-j*-pollen, leading to transmission ratio distortion. Knocking out one or two of the three *Sc-i* copies by CRISPR/Cas9 rescues *Sc-j* expression and male fertility. Our results reveal the gene dosage-dependent allelic suppression as a mechanism of hybrid incompatibility, and provide an effective approach to overcome the reproductive barrier for hybrid breeding.

[1] State Key Laboratory for Conservation and Utilization of Subtropical Agro-Bioresources, Guangzhou 510642, China. [2] Key Laboratory of Plant Functional Genomics and Biotechnology of Guangdong Provincial Higher Education Institutions, Guangzhou 510642, China. [3] College of Life Sciences, South China Agricultural University, Guangzhou 510642, China. [4] National Maize Improvement Center of China, Key Laboratory of Crop Genetic Improvement and Genome of Ministry of Agriculture, Beijing Key Laboratory of Crop Genetic Improvement, China Agricultural University, Beijing 100193, China. Correspondence and requests for materials should be addressed to Y.-G.L. (email: ygliu@scau.edu.cn)

Hybrid sterility (HS, including male and female sterilities) reflects the genetic incompatibility between evolutionarily divergent populations in the same or different species. This major form of postzygotic reproductive isolation restricts gene flow and maintains the separation of species (subspecies) during speciation[1–3]. The classical Bateson–Dobzhansky–Muller (BDM) model proposes that the hybrid incompatibility results from deleterious genetic interactions between divergent alleles[4–6]. Asian cultivated rice (*Oryza sativa* L.) was domesticated from the wild rice (*Oryza rufipogon* Griff.) and differentiated into two subspecies, *japonica* and *indica*[7]. Rice HS, as first reported in 1928, occurs extensively in the inter-subspecific (*japonica–indica*) hybrids, and is one of the major factors determining the genetic differentiation of these two subspecies[8].

**a**

T65 (*Sc-j/Sc-j*, FF)   F₁ (*Sc-j/Sc-i*, SS)   E5 (*Sc-i/Sc-i*, FF)

F₂, *Sc-j/Sc-j* : *Sc-j/Sc-i* : *Sc-i/Sc-i*
= 3:377:368 ($\chi^2$ = 356.3)

**b**

Marker   RG227   E14-E2

Chr. 3

*Sc* locus

Marker (kb): 114.0   100.7   93.0 92.0   90.5   85.7   84.4   80.4   38.0

No. recombinant: 4   2 0   0   0 2   5   BAC: OSJNBb0078P24
(from 8689 F₂ plants)

**c**

8.6 kb

*japonica* cv: T65, Nip, Koshihikari, Hitomebore, HEG4, A123

*Sc-j*
(*Os03g0247300*)

*indica* cv:

E5

14.9 kb   28.4 kb   28.4 kb   28.2 kb

*Sc-is*   *Sc-ib2*   *Sc-ib1*   23.9 kb   *Sc-ia*

T1   T2   23.9 kb   23.9 kb

Probe II   Probe I
(11.9 kb)   (16.3 kb)

MH63
93–11

*Sc-is*   *Sc-ib2*   *Sc-ib1*

**d**

MH63 BAC   Probe I (red)

*Sc-is*   28.4 kb (*Sc-ib2*)   28.4 kb (*Sc-ib1*)   2 µm

MH63 BAC   Probe II (red)

2 µm

E5 gDNA   Probe I (red) + Probe II (green)

*Sc-is*   28.4 kb (*Sc-ib2*)   28.4 kb (*Sc-ib1*)   28.2 kb (*Sc-ia*)   2 µm

The development and application of hybrid rice technology has contributed greatly to increase rice yield[9–11]. In the past decades, hybrid rice, which boosted grain yield by about 20% compared to inbred varieties, has been planted ca. 17 million hectares annually in China alone (accounting for about 60% of its total rice planting area)[10]. However, current hybrid rice varieties were bred mainly from crosses between lines from the same subspecies, mostly *indica* lines, and hybrid yields have reached a plateau due to the narrow genetic diversity of the parental lines, which results in lower degrees of hybrid vigor (heterosis). By contrast, the hybrids between *japonica* and *indica* cultivars have much stronger heterosis, thus holding great promise for further increasing yield potential[12].

In *japonica-indica* hybrids, the accumulated genetic effects of multiple HS loci cause very low pollen and spikelet fertilities, which prevents seed formation. Therefore, HS is a major barrier preventing the utilization of the stronger *japonica–indica* heterosis for achieving higher grain yield, and overcoming HS remains a major challenge in high-yield rice breeding. Genetic studies have identified approximately 50 loci involved in HS in rice[3], but the allelic relationships of some loci remain unclear. Two general models have been proposed to explain genetic HS in rice: the one-locus sporo–gametophytic interaction model and the duplicate gametic–lethal model[13–15]. The former model proposes that genetic interaction between divergent alleles of a single locus in the sporophytic cells causes abortion of gametes carrying a specific allele. In the later model, epistatic interactions between divergent alleles of two loci cause HS.

So far, the HS-causal genes have been cloned from five loci (*Sa*, *S5*, *HSA1*, *S7*, and the locus pair *DPL1/DPL2*) in *japonica–indica* hybrids[16–21], from a locus pair (*S27/S28*) in Asian rice–wild rice (*Oryza glumaepatula*) hybrids[22], and from a locus (*S1*) in hybrids of Asian rice and African rice[23]. *Sa* confers hybrid male sterility (HMS), and was identified as the first HS complex locus consisting of two adjacent genes *SaF* and *SaM*, which encode an F-box protein and a small ubiquitin-like modifier E3 ligase, respectively. In *Sa*-heterozygotes, detrimental interactions among proteins from three alleles (*indica* SaF and SaM, and *japonica* SaM) cause selective abortion of pollen grains carrying the *japonica* SaM allele, thus establishing the two-gene/three component interaction model at the molecular level[16]. *S5* also is a complex locus containing three tightly linked genes, *ORF3*, *ORF4*, and *ORF5*. During female sporogenesis, co-action of the killer/partner genes *ORF4* and *ORF5* of the *indica* allele causes endoplasmic reticulum stress and abortion of embryo sacs that contain the non-functional protector gene *ORF3* in the *japonica* allele; the functional *ORF3* of the *indica* allele rescues these effects[18]. Similarly, the *HSA1* locus contains two closely linked genes, *HSA1a* and *HSA1b*; epistatic interactions among the *japonica* and *indica* alleles of the genes in *japonica–indica* hybrids cause female gamete abortion[19]. At the *S7* locus, a gene encoding a tetra-tricopeptide repeat domain-containing protein confers the *Aus–japonica–indica* hybrid female sterility[20]. The locus pairs

*DPL1/DPL2* and *S27/S28* were generated from duplication of the pollen-essential genes on different chromosomes, followed by reciprocal loss of one of the duplicated genes in the divergent species; chromosomal recombination in male meiosis of the hybrids carrying *DPL1/DPL2* or *S27/S28* produces ~ 25% sterile pollen grains that lack both of the functional loci[21,22]. At the *S1* locus, a gene *OgTPR1* encoding a protein containing two trypsin-like peptidase domains and a ribosome biogenesis regulatory domain is required for the interspecific hybrid male and female sterilities[23]. Even though much has been learned about the genetic basis of HS in rice, we still lack a complete picture of its complexity and efficient methods to overcome HS for hybrid rice breeding.

Natural hybrid-compatible (neutral) alleles, which do not cause HS in hybrids, have been found for some HS loci (such as *S5-n* at *S5* and *Sa-n* at *Sa*) in rice[3,14–17]. However, previous efforts using neutral alleles to develop the inter-subspecific hybrids have had only limited success. In addition, breeding of hybrid wide-compatible lines with natural neutral alleles by successive back-crossing is time consuming, and often not feasible because no neutral alleles have been detected for many HS loci.

Genomic variations are the basis for genetic diversity, genome evolution, and natural selection in organisms. Genomic structural variations, including segmental insertion, deletion, inversion, translocation, duplication, and copy number variation (CNV), have significant genetic effects on various biological processes, agronomic traits, and human diseases[24,25]. For example, genomic CNVs increase nematode resistance in soybean[26] and contribute to grain-size diversity in rice[27]. However, how genomic structural variations generate speciation genes that contribute to reproductive isolation and regulate allele-specific gene expression and function remains largely unclear.

*Sc* is a major locus conferring HMS in *japonica–indica* hybrids and was identified using a cross between the *japonica* cultivar Taichung 65 (T65, carrying a *japonica* allele, *Sc-j*) and its near-isogenic line E5 (which has the T65 nuclear background but contains an *indica* allele, *Sc-i* introgressed from the *indica* cultivar Peh-ku)[13,28]. We previously mapped *Sc* to a region of chromosome 3 (ref. [29]). Here we report the isolation and characterization of the gene for *Sc*, and reveal that genomic structural changes and CNV at *Sc* result in allelic suppression that causes the hybrid incompatibility. Based on these findings, we also provide an effective approach that uses genome editing to break down the reproductive barrier for hybrid breeding.

## Results

**Identification of a DUF1618 protein gene at the *Sc* locus.** The F$_1$ hybrid (*Sc-j/Sc-i*) of the T65/E5 cross is partially (semi-) sterile due to the abortion of pollen grains carrying *Sc-j*, resulting in a severe, male-dependent transmission ratio distortion (mTRD) of the *Sc* alleles (Fig. 1a). Therefore, the *Sc-j* and *Sc-i* alleles are genetically incompatible in the inter-subspecific hybrids. So far,

**Fig. 1** Cloning of *Sc* reveals variations in genomic structure and copy number. **a** Pollen phenotypes of T65, E5, and their F$_1$. Arrows indicate smaller, sterile pollen grains (scale bars, 50 μm). The self-fertilized F$_2$ family shows a severely distorted segregation of the alleles ($P < 0.001$ for the chi-square test). **b** Fine-mapping of *Sc* using F$_2$ plants of the T65/E5 cross. The markers indicate the positions (kb) in the BAC of the *japonica* cultivar (cv) Nipponbare (Nip). **c** Genomic configurations of the *Sc* alleles in *japonica* lines (from GenBank except for T65) and *indica* lines (determined in this study). The *Sc-i* allele variants have three or two tandemly duplicated segments, each containing the gene homologous to *Os03g0247300*. T1 (2016 bp) and T2 (9087 bp) are transposon insertions (a DDE-type and a Rim2/Hipa-type, respectively). The *Sc-i* paralogous genes have recombinant promoter/upstream sequences different from that of *Sc-j*. **d** Fiber-FISH images of the MH63 BAC (~ 180 kb) and the E5 chromatin (genomic) DNA (gDNA). For the BAC fiber-FISH, Probes I and II were labeled for red fluorescence, and the whole BAC DNA was labeled for green fluorescence. The labeled Probe I (or Probe II) was mixed with the labeled BAC for the hybridization. For fiber-FISH of E5 gDNA, Probes I and II were labeled for red and green fluorescence, respectively, and mixed for hybridization. Scale bars, 2 μm. Note that the BAC and chromatin DNA fibers for the same segments had different extension levels, similar to the previous observations[34]. FF full fertility, SS semi-sterility

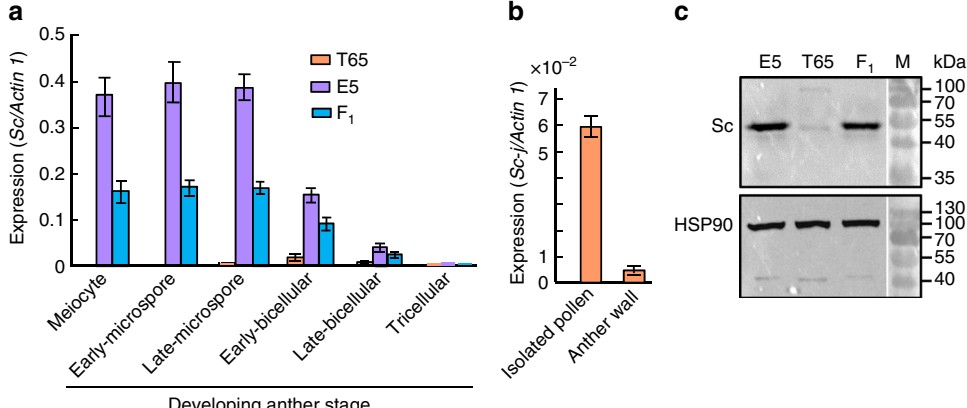

**Fig. 2** Expression of the *Sc* alleles. **a** Expression of *Sc-i* and *Sc-j* in the developing anthers of E5, T65, and their F₁ by qRT-PCR using a common primer pair (P3/P8, Supplementary Fig. 1) for all the alleles/*Sc-i* paralogs (except for *Sc-is*). The values (means of three biological replicates with s.e.m.) were normalized to *OsActin 1* transcript levels. **b** qRT-PCR analysis of *Sc-j* (using P3/P8) in separated pollen and anther wall cells (at early-bicellular stage) of T65. Data are shown as means ± s.e.m. (*n* = 3). **c** Different Sc protein abundances in anthers (at stages of uninucleate late-microspore to early-bicellular pollen) of the parents and the F₁, detected with anti-Sc antibodies

no neutral allele at the *Sc* locus has been identified in rice germplasm. By fine-mapping, using the mapping population derived from the T65/E5 cross, we delimited *Sc* to an 8.6-kb region (Fig. 1b). In *japonica* cultivars, this allele contains an annotated gene Os03g0247300, here named *Sc-j* (Fig. 1c). *Sc-j* encodes a DUF1618 domain-containing protein (446 amino acids, aa), belonging to a monocot-specific gene family[30].

**Complex genomic structural variations in the *Sc-i* alleles.** Analysis of the genome sequence at the *Sc* locus in the *indica* cultivar 93-11 (GenBank accession numbers: AAAA02008136.1, AAAA02008130.1, AAAA02008132.1, AAAA02008135.1) suggested complex structural variations at *Sc*, but the available sequence in this region was largely incomplete. We therefore sequenced a *Sc-i*-containing bacterial artificial chromosome (BAC) of the *indica* cultivar MH63, and PCR-amplified and sequenced the genomic and cDNA sequences of the *Sc-i* region in 93-11 and E5. The results showed that the complete genomic sequence of the *Sc-i* region in E5 contains four copies of the Os03g0247300 sequence (Fig. 1c and Supplementary Figs. 1, 2). One copy has a similar promoter sequence to *Sc-j* but is a pseudogene (named *Sc-is*) due to a 2-bp deletion and two transposon insertions (a DDE-type[31] and a Rim2/Hipa-type[32]) in this gene (Supplementary Fig. 2). The other three copies, named *Sc-ia*, *Sc-ib1*, and *Sc-ib2*, each contain the complete Os03g0247300-homologous sequence, and are located in three tandemly duplicated segments (28.2 or 28.4 kb in length) (Fig. 1c and Supplementary Figs. 1, 2). MH63 and 93-11 also have *Sc-is* and two duplications of the segment (28.4 kb) with *Sc-ib1* and *Sc-ib2*, but lack the *Sc-ia*-containing segment (Fig. 1c). Notably, the 23.9-kb upstream/promoter sequence of *Sc-ia*, *Sc-ib1*, and *Sc-ib2* differ completely from that of *Sc-j* (Fig. 1c and Supplementary Figs. 1, 2). The coding regions of *Sc-j* and these paralogous *Sc-i* genes include nineteen single-nucleotide polymorphisms (SNPs) and three 3-bp insertion/deletions, which result in several aa substitutions, but the coding sequences of *Sc-ib1* and *Sc-ib2* are identical (Supplementary Figs. 3 and 4). The *Sc-i* allele variants probably arose from genomic rearrangement between the ancestral DUF1618 gene and the 23.9-kb genomic sequence to produce a new recombinant gene, followed by inactivation of *Sc-is* and CNV of the ~ 28-kb segment.

Using PCR assays, we genotyped 14 additional *japonica* cultivars and 21 *indica* cultivars, and found that all tested *japonica* cultivars have the *Sc-j* allele and all *indica* cultivars have

the *Sc-i* allele with at least two duplications of the segment (Supplementary Fig. 5a–c). We further genotyped this locus in *O. rufipogon* and found that, of the nine accessions analyzed, five have the *Sc-i*-type structure (including *Sc-is* and at least two duplications of the ~ 28-kb segment) and four have the *Sc-j*-type structure (Supplementary Fig. 5d). This suggests that the divergence of *Sc* occurred before the domestication of *O. sativa* from *O. rufipogon* and the differentiation of the *japonica* and *indica* subspecies.

Since the PCR assay could not determine the actual copy number of the ~ 28-kb segment at the *Sc-i* allele variants, we further carried out DNA fiber-fluorescence in situ hybridization (fiber-FISH)[33,34] to examine the *Sc-i* structure of several *indica* cultivars. We found that five cultivars (MH63, 93-11, HHZ, Dular, and 02428) contain two duplicates (not including *Sc-is*) of the segment, whereas three other cultivars (E5, ZS97B, and GLA4) contain three such duplicates (Fig. 1d and Supplementary Fig. 6), consistent with the sequencing results of E5, MH63, and 93-11. These results indicate that the genomic structural change and the segment CNV at *Sc* is a common feature in *indica* cultivars.

**Different expression patterns of the *Sc-j* and *Sc-i* alleles.** We speculated that the structural variations might alter the expression pattern and/or level of the diverged *Sc* alleles; therefore, we examined their expression by semi-quantitative and quantitative reverse transcription PCR (qRT-PCR). *Sc-j* in T65 was expressed specifically in anthers, peaking but having a low level at the early-bicellular pollen stage (Fig. 2a and Supplementary Fig. 7a), which is consistent with the microarray-based transcriptome data from the *japonica* cultivar Nipponbare (Nip, with *Sc-j*) (Supplementary Fig. 7b). The repeated *Sc-i* paralogs in E5, however, were expressed broadly in vegetative organs and anthers of different stages, and the total expression level of all paralogs was much higher than that of *Sc-j* (Fig. 2a and Supplementary Fig. 7a). In the early-bicellular stage anthers, the *Sc-j* expression level (0.020, expressed as the ratio to *OsActin 1* expression) in T65 was only ~ 13% that of *Sc-i* (0.151) in E5 (Fig. 2a).

To verify if *Sc-j* is expressed in pollen, we carried out mRNA in situ hybridization to *Sc-j* in anthers of T65, but failed to obtain a clear signal, probably due to its very low-level expression. Alternatively, we separated pollen grains from the anther wall (epidermis) cells (at the early-bicellular stage) of T65, The results showed that *Sc-j* was expressed mainly in pollen and showed

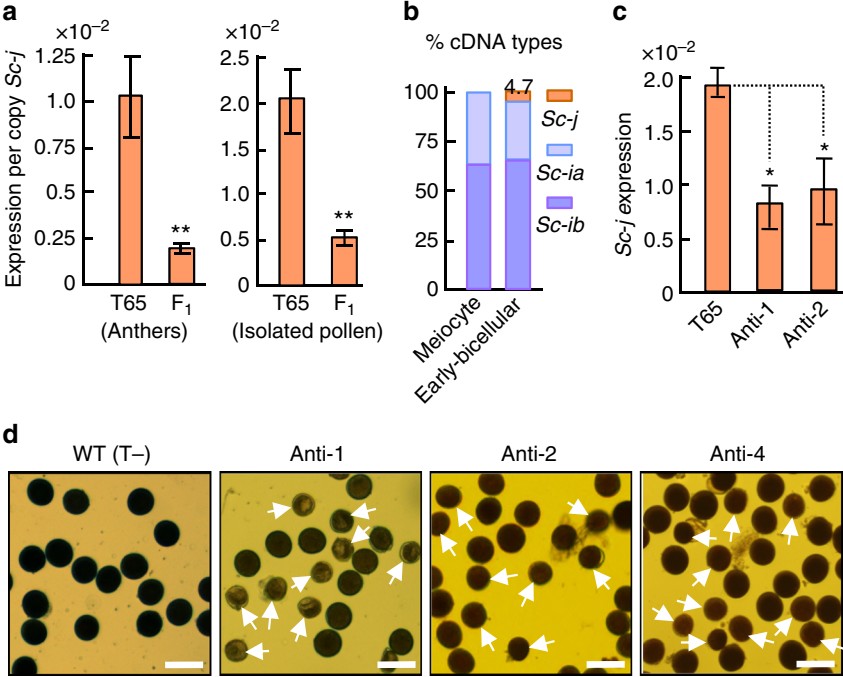

**Fig. 3** Functional analysis of *Sc-j*. **a** qRT-PCR analysis of relative *Sc-j* expression levels (using the *Sc-j*-specific primers P1/P2) in the anthers and isolated pollen (at early-bicellular stage) of the $F_1$ (T65/E5) and T65, calculated as the ratios of the transcript per copy *Sc-j* to the transcript of *OsActin 1* (two copies), because there is only one *Sc-j* copy in the $F_1$. The data are shown as means ± s.e.m. (*n* = 3), and * and ** indicate significance at *P* < 0.05 and *P* < 0.01, respectively, by two-tailed Student's *t* test (the same for Fig. 3c). **b** Percentages of cDNA clones from *Sc-j*, *Sc-ia*, and *Sc-ib* (*Sc-ib1/ib2*) of the $F_1$ anthers of two stages indicated. **c** Expression of *Sc-j* in the early-bicellular stage anthers of two $T_1$ (T65 background) plants carrying the hemizygous antisense-transgene of *Sc-j*. The detected transcript levels in Anti-1 and Anti-2 could be mainly from the pollen grains (~ 50%) without the antisense-transgene. **d** Semi-sterility of three $T_1$ plants carrying the hemizygous antisense-transgene. WT (T-), a $T_1$ segregant without the antisense-transgene. Scale bars, 50 μm

much less expression in the anther wall (Fig. 2b), consistent with its proposed function in pollen development (see below). The differential expression of the *Sc-j* and *Sc-i* alleles was also reflected by their different protein abundances (Fig. 2c and Supplementary Fig. 8). We speculated that the high-level and broad expression patterns of *Sc-i* could result from the repeated gene copies and the distinct promoter sequence.

**The expression of *Sc-j* is largely suppressed in the hybrid**. To explore the possible causal relationship between the differential expression of the *Sc* alleles and HMS in the hybrid, we examined *Sc-j* expression levels in the hybrid and T65 using *Sc-j*-specific primers. In the early-bicellular stage anthers, the expression level of *Sc-j* per copy in the hybrid was about 1/5 of that in T65 (Fig. 3a). Similar results were obtained in isolated early-bicellular stage pollen grains (Fig. 3a), suggesting that *Sc-j* expression is largely repressed in the hybrid pollen grains. We further analyzed the relative transcript abundances of *Sc-j* and the *Sc-i* paralogs in the hybrid by genotyping cDNA clones based on the SNPs in *Sc-j* and the *Sc-i* paralogs (Supplementary Fig. 3). In anthers of the meiocyte (microspore mother cell and meiosis I) and uninucleate microspore stages, no or very few *Sc-j*-type cDNA clones were detected; in the early-bicellular stage anthers, *Sc-j* type cDNAs accounted for only 4.7% of the total clones (Fig. 3b and Supplementary Table 1), much lower than the expression level of *Sc-j* in T65 (13%) compared to E5 (Fig. 2a). Together, these results suggest that normal pollen development requires a threshold level of *Sc-j* expression in the early-bicellular pollen, and that the suppression of expression *Sc-j* in the hybrid, a phenomenon that we term "allelic suppression", might be the cause of *Sc-j*-pollen abortion.

Previous cytological observation of sectioning pollen of the T65/E5 hybrid showed that at the bicellular stage, the cell wall (callose) of the mitosis I-generated generative cell (nucleus) within *Sc-j*-pollen failed to degenerate at the proper time and the generative nucleus could not move with the vegetative nucleus as occurs in normal bicellular pollen. This leads to the generative nucleus being unable to complete mitosis II[35]. Therefore, this allelic suppression of *Sc-j* in early-bicellular pollen may be related to the cytological mechanism of impaired *Sc-j*-pollen development.

**Sc is a gametophytic factor for pollen development**. To further test if *Sc* is essential for pollen development, we introduced an *Sc*-antisense construct into T65 plants. As expected, the hemizygous transgenic plants with reduced *Sc-j* expression in their anthers showed partial male-sterility, similar to the *Sc*-mediated HMS (Fig. 3c, d). Furthermore, the transgene segregation ratio in five $T_1$ families (from self-fertilization) fitted the 1:1 ratio expected for a hemizygous gametophytic sterility gene (Supplementary Table 2). Therefore, we concluded that this DUF1618 gene has a primary function as a gametophytic factor, and normal pollen development requires a threshold level of *Sc* expression.

To test whether the abnormal development of *Sc-j*-pollen involves changes in the expression of genes known to affect male fertility[36,37], we analyzed a microarray-based transcriptome data set from anthers of T65 and the T65/E5 $F_1$. Of 14 genes with known functions in microspore/pollen development, four (*OsADF*, *OsGSL5*, *OsUgp2*, and *OsCAP1*) showed differential microarray hybridization signals (Supplementary Fig. 9a). However, further qRT-PCR analysis did not find significant difference in expression of these four genes in developing anthers between T65 and the $F_1$ (Supplementary Fig. 9b).

**Table 1 Segregation distortion in various crosses using *indica* parents with different copy numbers of *Sc-i***

| Cross ♀/♂ | F₁ genotype | No. plant (F₂) | Segregation *jj:ji:ii* | % *jj* plant | $\chi^2$ (1:2:1) |
|---|---|---|---|---|---|
| E5/T65 | *Sc-j/3×Sc-i* | 748 | 3:377:368 | 0.4 | 356.3*** |
| ZS97B/Nip | *Sc-j/3×Sc-i* | 213 | 7:118:88 | 3.3 | 64.1*** |
| Nip/HHZ | *Sc-j/2×Sc-i* | 134 | 6:65:57 | 4.7 | 40.7*** |
| Nip/93-11 | *Sc-j/2×Sc-i* | 258 | 25:145:88 | 9.7 | 34.7*** |
| Nip/02428 | *Sc-j/2×Sc-i* | 172 | 12:95:65 | 7.0 | 34.6*** |
| | | (RIL) | *jj:ii* | | (1:1) |
| GLA4/Nip | *Sc-j/3×Sc-i* | 106 (F₁₀) | 1:105 | 0.9 | 102.0*** |
| Nip/93-11 | *Sc-j/2×Sc-i* | 247 (F₇) | 52:195 | 21.1 | 82.8*** |

Note: The *Sc-i* allele variants have three (3×) or two (2×) segment copies (Fig. 1d and Supplementary Fig. 6). *jj*, *ij*, and *ii* indicate homozygous *Sc-j*, the heterozygote, and homozygous *Sc-i*, respectively. ***significance at $P < 0.001$ for the $\chi^2$ test. The expected normal frequencies of plants with *jj* genotype are 25% for F₂ and 50% for RIL. All crosses with the *japonica* cultivars as the male or female parents produced lower transmission rates of *Sc-j*

**CNV of *Sc-i* is associated with the degree of HMS.** To investigate the relationship between the *Sc-i* CNV and HMS, we crossed the *japonica* cultivar Nip with *indica* cultivars that have three copies (E5 and ZS97B) or two copies (93-11, HHZ, and 02428) of the ~ 28-kb segments (Supplementary Fig. 6). The two F₂ families with three *Sc-i* copies had more severe mTRD (with 0.4% and 3.3% of *Sc-j* homozygotes) than those of the three F₂ families with two *Sc-i* copies (with 4.7–9.7% of *Sc-j* homozygotes) (Table 1). Particularly, the recombinant inbred lines (RILs) derived from crossing Nip with the *indica* cultivar GLA4 (carrying three *Sc-i* copies) produced only 0.9% *Sc-j* homozygotes, whereas the frequency was 21.1% in the RILs derived from crossing Nip with 93-11. The more severe mTRD reflects greater abortion of *Sc-j*-pollen in the *Sc*-heterozygotes. These results suggest that the severity of HMS in the hybrids is associated with the CNV at *Sc-i*. Notably, all the *indica* cultivars analyzed have at least two *Sc-i* copies (Supplementary Figs 5 and 6) and these *Sc-i* allele types produced different degrees of genetic incompatibility to *japonica* cultivars (Table 1).

**Reduction of the *Sc-i* dosage rescues the HMS.** Based on the relationship of the *Sc-i* CNV with the HMS, we tested whether decreasing the number of functional *Sc-i* copies by genome editing could generate artificial neutral alleles. Using our CRISPR/Cas9 plant genome editing system[38], we made two binary constructs, one to delete the ~ 28-kb segment(s) and another to specifically edit a unique target site in *Sc-ib1* and *Sc-ib2* (but not in *Sc-ia*) based on a SNP that forms the protospacer adjacent motif (PAM) required for CRISPR/Cas9 genome editing (Fig. 4a). By transformation of E5 with these constructs, we identified a plant (E5-d1) with one of the segmental duplications deleted (Fig. 4a and Supplementary Fig. 6), and two plants (E5-ed1 and E5-ed2) with nucleotide mutations in *Sc-ib1* and *Sc-ib2*, but with their *Sc-ia* intact due to the lack of a PAM (Fig. 4a). We crossed these edited plants with T65 to obtain F₁ plants (F₁-d1, F₁-ed1, and F₁-ed2) and confirmed the edited sites by sequencing (Supplementary Fig. 10). The pollen fertility of these hybrids was greatly improved (Fig. 4b) and the F₂ families showed less mTRD; two F₂ families from F₁-ed1 and F₁-ed2 had segregation ratios fitting the 1:2:1 ratio (Table 2), thus confirming the restoration of *Sc-j*-pollen fertility. Therefore, reduction of the gene dosage by the targeted mutation of two of the three *Sc-i* paralogs can convert the hybrid-incompatible *Sc-i* allele into completely compatible alleles while maintaining the basic function of *Sc* in gametogenesis. We name these edited *Sc-i* alleles *Sc-n*, as they produce no significant HMS.

To verify the causal relationship between expression of *Sc-i* and *Sc-j* and male fertility in the edited hybrids, we analyzed the mRNA levels of *Sc-ia*, the mutated *Sc-ib1* and *Sc-ib2*, and *Sc-j* in anthers of the edited F₁ plants using the intact and edited allele-specific primers (Fig. 4a). The mutated mRNA levels were greatly reduced relative to the *Sc-ia* transcript (Fig. 4c), probably due to the common nonsense-mediated mRNA decay mechanism for abnormal mRNAs[39]. In contrast, the *Sc-j* transcript levels (at the early-bicellular stage) of these edited F₁ plants significantly increased (Fig. 4d). Thus, the *Sc-j* expression level and the pollen fertility in the *Sc*-heterozygotes are negatively associated with the *Sc-i* gene dosage and its expression level.

**Discussion**

In this study, we isolated and characterized the gene at the HMS locus *Sc*. Based on the results, we propose a molecular genetic model for the *Sc*-HMS system. In the wild rice, the pollen-essential *Sc* gene underwent structural changes and CNV, and the divergent *Sc-j*-type and *Sc-i*-type alleles were inherited into the *japonica* and *indica* subspecies, respectively, during domestication. In the *japonica*–*indica* hybrids, the *Sc-i* paralogs are highly expressed in the sporophytic cells and the product may interact, directly or indirectly, with the promoter region of *Sc-j* by a yet-unknown mechanism to mediate the suppression of *Sc-j* expression. This interaction may involve transcriptional-repressive epigenetic modification(s) of the *Sc-j* promoter region, and the modification state (epiallele) may be retained, through meiosis, in the pollen grains, thus leading to the suppression of *Sc-j* transcription in pollen and the allele-specific male-gamete elimination (Fig. 5). This model of gene dosage-dependent HMS is consistent with the partial, sometimes almost complete, restoration of hybrid male fertility produced by reducing the number of copies in the *Sc-i* locus in the natural allele variants and the genome-edited alleles of *Sc-i*.

The change of the promoter sequence of *Sc-i* via structural variations not only alters its gene expression pattern and levels, producing the putative allelic interaction in the sporophytic cells, but might also allow the *Sc-i* paralogs to escape repression by their own products. Thus, the *Sc-i* allele variants have evolved into selfish speciation genes in this HMS system, and can cause specific abortion of pollen with *Sc-j* but not pollen carrying the *Sc-i* variants in the hybrids. Such a scenario exemplifies the molecular genetic basis of the classical one-locus sporo-gameto-phytic interaction model for plant HS[13–15]. The BDM-type genetic incompatibility results from the divergence and interaction of genetic loci during evolution. It would be interesting to know if other intermediate types of *Sc-i*-like and/or *Sc-j*-like allele variants that are likely hybrid-neutral or hybrid-incompatible exist in wild and cultivated rice populations, for example, the primarily diverged *Sc-i*-like allele having a novel, recombinant promoter but without the segmental duplication. Therefore, the *Sc*-HMS system may be an ideal model for study of BDM-type or a novel type of postzygotic hybrid incompatibility caused by a single locus. Further investigation will be required to analyze the

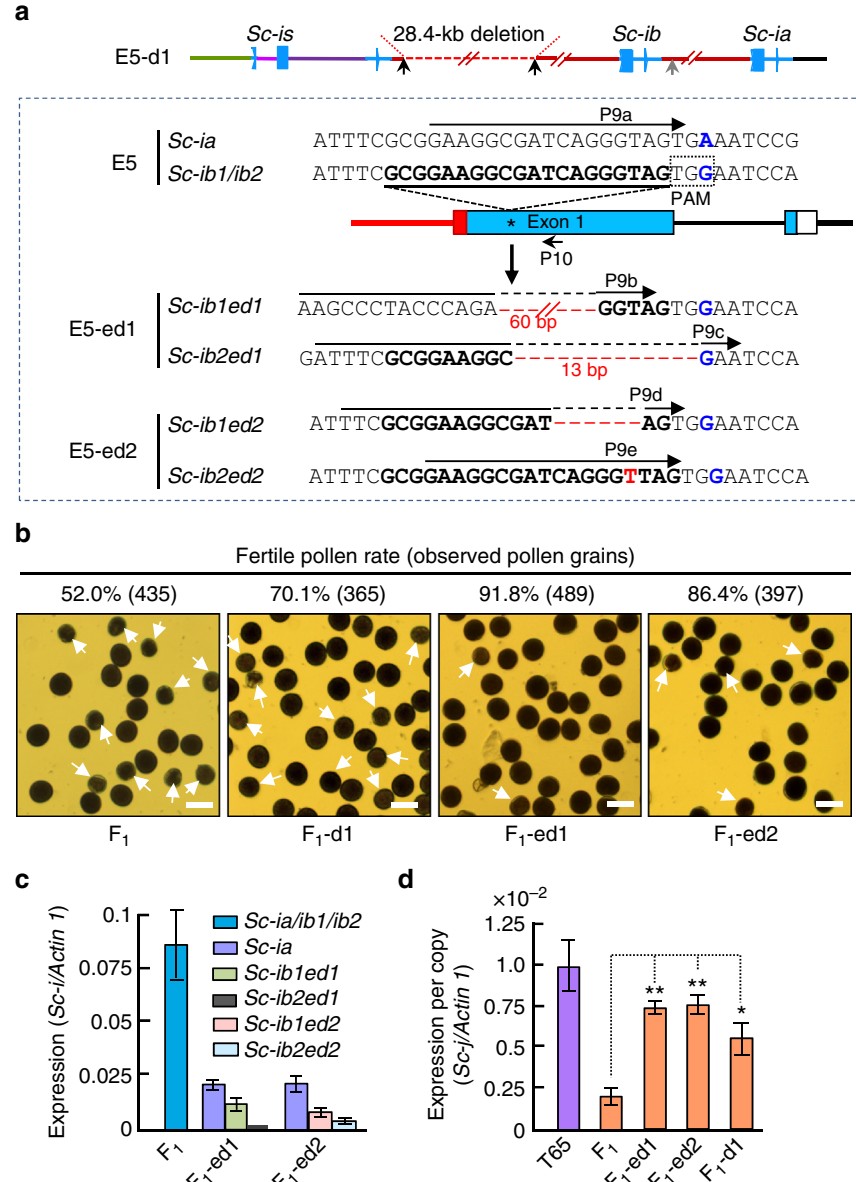

**Fig. 4** Genomic editing of the *Sc-i* paralogs rescues *Sc-j* expression and male fertility in the hybrids. **a** Targeted editing of the sites (arrowed) in E5 produced a plant (E5-d1) with deletion of one of the ~ 28-kb segments (Supplementary Fig. 6), and editing another site (underlined) specific to *Sc-ib1* and *Sc-ib2* of E5 identified two plants (E5-ed1 and E5-ed2) with mutated *Sc-ib1* and *Sc-ib2* and intact *Sc-ia* (Supplementary Fig. 10). The SNP (blue) in *Sc-ib1* and *Sc-ib2* formed the PAM (NGG) required for genome editing. Primers P9a-P9e combined with the *Sc-i*-specific P10 were used for qRT-PCR of the intact and mutated *Sc-i* transcripts show in (**c**). **b** Improvement of pollen fertility (sterile pollen grains arrowed) in the edited $F_1$ plants from crossing E5-d1, E5-ed1, and E5-ed2 with T65. Scale bars, 50 μm. **c** Expression levels (in the meiocyte stage anthers) of the *Sc-i* paralogs in the $F_1$ (T65/E5), and the intact *Sc-ia* and mutated *Sc-ib1*/*Sc-ib2* in the edited $F_1$ plants. Data are shown as means ± s.e.m. (*n* = 3). **d** Expression levels of per copy *Sc-j* (qRT-PCR using P1/P2) in the early-bicellular stage anthers of the intact and edited $F_1$ plants and T65. The data are shown as means ± s.e.m. (*n* = 3), and * and ** indicate significance at $P < 0.05$ and $P < 0.01$, respectively, by two-tailed Student's *t* test

**Table 2 *Sc* genotypic segregation in the edited $F_2$ families**

| $F_2$ family | No. plant | Segregation *jj:ji:ii* | % *jj* plant | $\chi^2$ (1:2:1) |
|---|---|---|---|---|
| $F_2$ | 176 | 1:93:82 | 0.6 | 75.1*** |
| $F_2$-d1 | 262 | 32:141:89 | 8.4 | 26.3** |
| $F_2$-ed1 | 256 | 54:133:69 | 20.5 | 2.1 |
| $F_2$-ed2 | 199 | 36:107:56 | 18.1 | 5.15 |

Note: *jj*, *ii*, and *ji* denote homozygous *Sc-j*, homozygous *Sc-i* (intact or edited), and the heterozygote, respectively. ** and ***, significance at $P < 0.01$ and $P < 0.001$, respectively, for the $\chi^2$ test

allelic interaction and the molecular mechanism of *Sc*, as well as its evolution.

The development of anthers and pollen involves a large number of genes, and many of them are highly conserved between monocot and dicot plant species[36,37]. However, this study revealed that the *Sc* gene is a member of the monocot-specific DUF1618 gene family (121 members in *japonica* rice). To date, only two genes of this family in rice, *HSA1a*[19] and *Sc*, have been functionally studied. These studies showed that *HSA1a* and *Sc* have important roles in reproductive development and participate in hybrid female and male sterilities, respectively. These

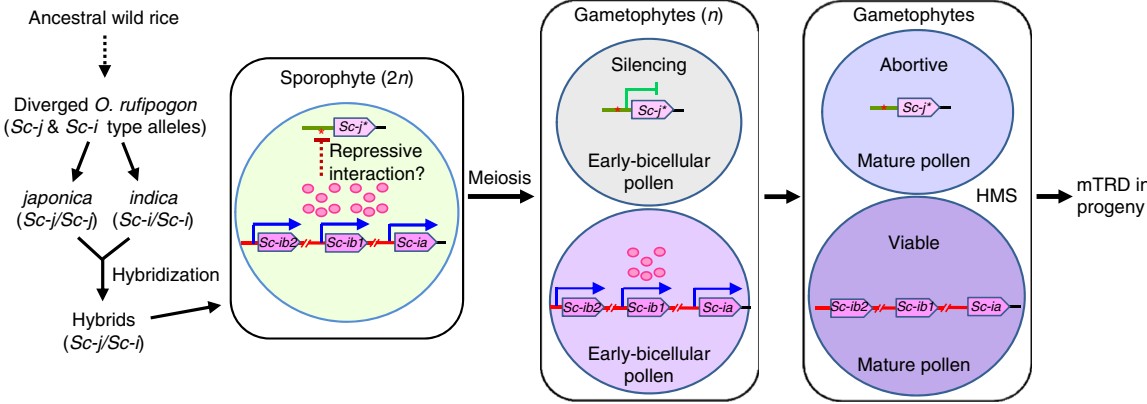

**Fig. 5** A proposed model for the molecular genetic mechanism of *Sc*-mediated HMS. In a *japonica–indica* hybrid, the *Sc-i* paralogs are highly expressed in sporophytic cells and the product may interact with the *Sc-j* promoter region. This might cause transcriptional-repressive epigenetic modification(s) (red asterisk) in the *Sc-j* promoter region. This putative modification state (*Sc-j\**) could be retained, through meiosis, in the male gametophytes, thus causing allelic suppression of *Sc-j* expression and the consequent *Sc-j*-specific pollen abortion (HMS) and mTRD. Only two of the four pollen grains from a meiocyte, and one *Sc-j* copy and one set of *Sc-i* copies in the bicellular and tricellular pollen grains, are shown

findings demonstrate that rice has evolved genetic networks distinct from those of dicots for male and female gametogenesis.

The findings from studies of a number of cloned HS loci in plants suggest that hybrid incompatibility involves diverse molecular genetic mechanisms[40]; the only common feature among some hybrid incompatibility systems is that they are caused by gene duplication/reciprocal loss-of-function mutations, as shown in the *DPL1/DPL2* and *S27/28* locus pairs[21,22]. In this study we revealed a new type of combinatorial genomic variation involving structural changes and CNV, and demonstrate that this type has an important genetic effect on hybrid incompatibility. Our results show that the *Sc-j* and *Sc-i* alleles are associated with *japonica* and *indica* cultivars, suggesting that this hybrid incompatibility might have contributed to the differentiation of the subspecies, although extensive analysis will be required to resolve this issue. The allelic suppression described here suggests a regulatory mode for differential allelic expression, which is characterized by allele-specific (*Sc-j*) repression and is driven by the opposite allele (*Sc-i*) of the same gene. This phenomenon has features distinct from the previously described paramutations[41] and monoallelic expression forms (parent-of-origin imprinting, random monoallelic expression of autosomal genes, and X-inactivation genes)[42–44].

In this study, we used CRISPR/Cas9 genome-editing technology to create neutral alleles of *Sc* by precisely reducing the gene dosage of *Sc-i* but maintaining its basic function in pollen development. By contrast, precise down-regulation of target genes remains difficult to achieve with other methods, such as antisense and RNA interference. The genome editing-based approach can rapidly generate neutral alleles at *Sc* and other HS loci that require loss-of-function mutations to generate neutral alleles, such as *Sa*, *S5*, and *S1*[16,17,23,45]. This may enable breeding of wide-compatible lines to break down the reproductive barrier and overcome HS in the inter-subspecific and inter-specific hybrid breeding in crops, and thus may have significant implications for agriculture.

## Methods
**Rice materials**. Rice materials for the gene cloning and functional analysis included the *japonica* line T65 and its near-isogenic line E5[13,28] and their hybrid progeny prepared in this study. The other *indica* cultivars MH63, ZS97B, HHZ, and GLA4 are Minghui 63, Zhengshan 97B, Huanghuazhan, and Guangluai 4, respectively. The line 02428 with a majority *japonica* background was derived from a *japonica–indica* cross[46]. The wild and cultivated rice species used for the haplotype survey and crosses were maintained in our laboratory. The rice plants were planted in experimental fields or in a phytotron (for some seedlings) at the South China Agricultural University in Guangzhou.

**Map-based cloning**. $F_2$ plants of the T65/E5 cross were used for mapping. The pollen phenotypes (male fertile or male sterile) of rice were observed by placing pollen in a 1% $I_2$-KI (potassium iodide) solution and observing the pollen under a microscope (Leica DMLB, Zeiss Axio Observer D1). Plants homozygous for *Sc* produced fully fertile pollen (>95% pollen fertile); plants heterozygous for *Sc* produced semi-fertile pollen (ca. 50% of pollen grains sterile). A number of insertion/deletion markers (Supplementary Table 3) were used for the fine-mapping. The key recombinant plants (Fig. 1b) were further confirmed by analyzing the phenotypic and genotypic segregation in their $F_3$ progenies. Genetic segregation of the genotypes in various populations was analyzed by $\chi^2$ test.

**Sequence analysis**. The BAC clone (OSIMNBa0008K19) was from an MH63 genomic library[47]. With the incomplete genomic sequences of the *indica* 93-11 at the *Sc* region (GenBank accession numbers: AAAA02008136.1, AAAA02008130.1, AAAA02008132.1, AAAA02008135.1) as references, a number of primer pairs were prepared and long-length PCR with KOD FX DNA polymerase (TOYOBO, Japan) was used to amplify the *Sc-i* region sequences from the BAC and from gDNAs of E5 and 93-11 in multiple fragments. The PCR products were sub-cloned into a plasmid vector and multiple clones for each fragment were sequenced. The sequences were assembled using SeqMan of the Lasergene package, and analyzed by BLAST (http://www.ncbi.nlm.nih.gov/BLAST/). The DDE-type and Rim2/Hipa-type transposon insertions in *Sc-is* were analyzed by BLAST (http://www.ncbi.nlm.nih.gov/BLAST/).

**DNA fiber-FISH**. For BAC fiber-FISH, the BAC plasmid was isolated from 200 ml LB culture using the alkaline lysis method. The BAC DNA was diluted to about 10 ng/µl in AF solution (100% glycerin:1 M sodium chloride:double-distilled water, 180:3:17, v/v, pH 7.5) with cut pipet tips. The diluted BAC DNA (2 µL) was added to Poly-Prep slides (Sigma-Aldrich, USA), then the slides were slowly covered with coverslips (22 × 22 mm) and set vertically to stretch the BAC DNA. The slides were air-dried for 10 min, then carefully uncovered by removing the coverslips. The slides were fixed in fresh Carnoy's solution (ethanol:glacial acetic acid, 3:1, v/v). Probe I (16.3 kb) and Probe 2 (11.9 kb) DNAs were amplified by PCR in three and two segments, respectively (primer sequences are provided in Supplementary Table 3), and purified on a 1% agarose gel with a gel-extraction kit. According to the DNA fiber-FISH protocol[33], the probe sequences (Probe I and Probe II) were labeled by a nick translation reaction with digoxigenin-11-dUTP (Roche, Switzerland), and mixed with the whole BAC DNA labeled with biotin-16-dUTP (Roche, Switzerland) for two separate hybridizations. The probes were firstly added to the slides and covered with the cover slips. Then the slides were denatured at 80 °C in for 5 min in an oven. Then the slides were immediately transferred to another oven and incubated at 37 °C overnight. Immunodetection of the probes were performed by sequentially adding various antibodies to gain amplifying the fluorescence signals. The biotin-labeled probes were detected successively with fluorescein-avidin D (Vector, USA), biotinylated anti-avidin antibody (Vector, USA), and fluorescein-avidin D (Vector, USA). The digoxigenin-labeled probes were firstly detected by anti-digoxigenin-rhodamine (Roche, Switzerland), then by texas red® goat anti-rabbit IgG antibody (Vector, USA). The slides were counterstained with 4′,6-diamidino-2-phenylindole (DAPI) in vecta-shield mounting medium (Vector, USA).

For genomic DNA fiber-FISH, nuclei were prepared from young rice seedling leaves, using the plant nuclei isolation method[48]. The seedling leaves were grounded to powder in liquid nitrogen, and the powder was suspended in buffer (10 mM Tris–HCl (pH 9.5), 10 mM EDTA (pH 8.0), 100 mM KCl, 0.5 M sucrose, 4 mM spermidine, 1 mM spermine, 0.1% mercaptoethanol). After filtering sequentially through 149, 74 and 37 μm nylon meshes, the nuclei were collected by centrifuging at 1500 g for 10 min. The genomic DNA fiber-FISH was performed following the protocol[34]. Briefly, the DNA fibers were extended by dragging with a cover slip upon the Poly-L-lysine (Sigma-Aldrich, USA) slides to get long and clean DNA fibers. The slides were hybridized using mixed Probe I labeled with digoxigenin-11-dUTP and Probe II labeled with biotin-16-UTP. The following procedures of hybridization and probes detection were carried as the same as the above description for BAC fiber-FISH. The fiber-FISH images of the BAC and genomic DNAs were captured digitally using a CCD camera (QIMAGING, USA) attached to an Olympus BX61 epifluorescence microscope. Gray-scale images were captured for each color channel and then merged. The final image optimization was performed using Adobe Photoshop CS3 (Adobe, USA).

**Immunoblot analysis**. A peptide antigen (MVEGNEGEESTKRMKVK) corresponding to amino acid residues 427–443 of Sc-i (430–446 of Sc-j) was synthesized and used for immunizing rabbits to produce the anti-Sc antibodies by Invitrogen (USA) as a customary service.

Detection of Sc-j and Sc-i with the antiserum was carried out as described with some modifications[49]. About 50 mg anthers (at stages of late-microspore to early-bicellular pollen) of T65, E5, and their hybrid F$_1$ were grinded to fine powder in liquid nitrogen and lysed in 500 μl 2 × protein extraction buffer (100 mM Tris-HCl (pH 7.5), 300 mM NaCl, 2 mM EDTA, 10% glycerol, 0.5% TritonX-100 and protease inhibitor cocktail from Roche, PMSF from AMRESCO). After centrifugation at 12,000 g for 15 min, the supernatants were collected and the total proteins were quantified with the 2-D Quant Kit (GE Healthcare, USA). Extracted proteins were mixed with an equal volume of the 2 × loading buffer (100 mM Tris-HCl, 5 mM DTT, 4% SDS, 0.01% bromophenol blue and 30% glycerol, pH 6.8) and boiled for 5 min, then separated by 12% SDS-PAGE and transferred onto an Immobilon-PPSQP transfer membrane (polyvinylidene fluoride (PVDF) type, Millipore) using a Bio-Rad mini-transfer cell. The membranes were incubated in blocking buffer (5% milk, 0.1% Tween-20, 0.1% Triton X-100, 100 mM Tris-HCl and 150 mM NaCl, pH 7.5) for 2 h at room temperature, washed twice with 2 × TBST buffer (0.1% Tween-20, 100 mM Tris-HCl and 150 mM NaCl, pH 7.5). Then the membranes were incubated with anti-Sc polyclonal antibodies (1:2000 dilution) or the anti-Hsp90 monoclonal antibody (Beijing Protein Innovation, China; AbM51099-31-PU) diluted (1:10000) in blocking buffer for about 20 h at 4 °C. After washing three times with TBST for 5 min each, the membranes were incubated in the secondary antibodies, HRP-conjugated goat anti-rabbit IgG or HRP-conjugated goat anti-mouse IgG (TransGen Biotech, China), with 1:10000 dilution for 1 h at room temperature and washed three times with TBST for 5 min each. The membrane blots were incubated in the ECL substrate (Bio-rad, USA) for 1 min. The photos were captured using Gel Logic 6000 Pro (Kodak, USA) and processed with Adobe Photoshop CS3 (Adobe, USA).

**Expression analysis**. Separation of pollen grains from anther wall cells was carried out by filtering broken anthers in 2% sucrose through a nylon mesh (48 μm in diameter), followed by centrifugation at 250 g for 10 min to pellet pollen. The separated pollen grains and anther wall cells were collected for RNA preparation. Total RNA was extracted from rice tissues (anthers, isolated pollen, anther wall cells, young panicles, leaves, stems, and roots) of T65, E5, and their hybrid using TRIzol reagent (Invitrogen, USA). The RNA samples were treated with DNase I (Promega, USA). First-strand cDNA was synthesized from 2 ~ 3 μg total RNA using Superscript III reverse transcriptase (Invitrogen, USA) with oligo dT or a Sc-specific primer. The expressions of Sc and the pollen-essential marker genes were investigated by qRT-PCR analysis with OsActin 1 (Os03g0718100) for normalization by three biological replicates, and the significance was analyzed by two-tailed Student's t test. SYBR Green QPCR mix (Bio-Rad) was used for qRT-PCR. The reactions were carried out using the Bio-rad CFX Connect Real-Time PCR System with the following profile: 95 °C for 3 min; 42 cycles of 95 °C for 10 s, 60 °C for 15 s, 72 °C for 20 s. The expression pattern of Sc-j (Os03g0247300) in Nip was from the Rice Expression Profile Database (http://ricexpro.dna.affrc.go.jp/)[50].

For precise quantification of the ratio of relative Sc-j and Sc-i copy numbers and transcript levels in the F$_1$, the gDNA and cDNA segments that included all the alleles/paralogs in different stage anthers of the F$_1$ plants were amplified using primers P5/P7. The PCR products were cloned into a plasmid vector, and multiple clones (196–380) for each sample were genotyped. The insertion/deletion marker 3-N1/3-N2 (Supplementary Table 3) was used to differentiate Sc-j from Sc-ia/Sc-ib1/Sc-ib2, and the SNP marker HRM1F/HRM1R to differentiate Sc-ia from Sc-ib1/Sc-ib2 (Supplementary Fig. 8). SNP genotype was detected by high-resolution melting analysis (LightScanner, USA) using EvaGreen (Biotium, USA) as a fluorescent dye[51].

**Constructs for rice transformation**. A 771-bp fragment covering the 3' coding region of Sc-j (from +514 to +1284 bp) was amplified and inserted in reverse orientation into an overexpression vector with the maize Ubiquitin promoter to prepare an antisense expression construct. Two genome-targeting constructs for fragmental deletion (the target site: 5'-GGCACAATGAGCGGGCTGGATGG-3', PAM underlined) and Sc-ib1/ib2 mutation (Fig. 4a) were prepared using the CRISPR/Cas9 vector system[38]. These target site sequences were searched against the rice genome sequence with the web-based tool CRISPR-GE/targetDesign (http://skl.scau.edu.cn/)[52] to confirm its specificity in the genome. The target-adapter primers are listed in Supplementary Table 3. All constructs were introduced into Agrobacterium tumefaciens strain EHA105 by electroporation and used for rice transformation. The PCR amplicons containing the target sites were directly sequenced and cloned in a plasmid for sequencing. The superimposed chromatograms of the direct sequencing were decoded with CRISPR-GE/DSDecodeM[52,53], and the PCR products also were cloned in a plasmid vector and multiple clones were sequenced.

**Data availability**. The nucleotide sequence data for Sc-j of T65 and the Sc-i variants of E5 and MH63 have been deposited in GenBank nucleotide database under accession numbers MF370526, KX495643 and KX495644, respectively. All the relevant data are available from the authors upon request.

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

## Acknowledgements

We thank G. Zhang, C. Zhuang and X.D. Liu (South China Agricultural University, SCAU) for providing T65, E5, and some other rice materials. We also thank H. Wang (SCAU), D. Charlesworth (University of Edinburgh), K. Tsunewaki (Kyoto University), H. Ma (Fudan University), Q. Zhang and Y. Ouyang (Huazhong Agricultural University) for comments on the manuscript. This work was supported by grants from the Ministry of Science and Technology of China (2013CBA01401, 2013CB126904 and 2012AA10A303), the National Nature Science Foundation of China (31401449), and the Nature Science Foundation of Guangdong Province, China (S2012040007477).

## Author contributions

R.S. performed most of the experiments; L.W., X.L., J.W., W.J., X.Z., X.X., Q.Z., H.T., Q.L., and L.C. performed some of the experiments. R.S. and Y.-G.L. analyzed tne data. Y.-G.L. conceived and supervised the project, and Y.-G.L. and R.S. wrote the manuscript.

## Additional information

**Competing interests:** The authors declare no competing financial interests.

