## [Peer Review File · Nature Communications]

Reviewers' comments:

Reviewer #1 (Remarks to the Author):

Hybrid sterility is frequently observed in hybrids of japonica and indica rice subspecies. Shen et al., reported that the genomic structural change and copy number variation of Sc locus are responsible for japonica-indica hybrid male sterility. The japonica allele Sc-j encodes a DUF1618-domain protein which is required for normal pollen development, whereas the counterpart of indica Sc-i allele has two or three tandem-duplicated ~28-kb segments, and each containing a Sc-j-homolog but with a varied and recombinant promoter. The author observed that high-expression of the Sc-i copies in hybrid anthers result in the suppressed expression of Sc-j, causing aborted pollen development. The authors also validated this conclusion by that the transgenic plants of knock-outing three Sc-i copies using CRISPR/Cas9 editing have the recovered expression of Sc-j and male fertility. This work provides insight into hybrid incompatibility in plants, will be useful for overcoming the reproductive barrier for crop breeding.

- 1)The authors investigate the expression of Sc using PCR in anthers. I suggest to check the expression in other tissues and in situ analysis is suggested for testing which cell type Sc expressed. The authors claimed that the Sc gene is a gametophytic factor for pollen development, it is essential to know the detailed cell type of the gene expression.
- 2)The authors claimed that Sc is required for pollen formation. But the evidence on the role of Sc in pollen development is not enough. I suggest the authors to check the anther developmental changes in the male fertile and sterile lines by sectioning anthers and pollen grains.
- 3)To confirm the function of Sc in pollen development, is it possible to use genome editing method instead of RNAi approach?
- 4)To understand the mechanism on how the hybrid is male sterile, is it possible to checked the expression of the genes known to be required rice male development (reviewed by Zhang and Liang Improving food security: Using male fertility for hybrid seed breeding. Science. 2016, sponsored collection: 45; Shi et al.,. Trends in Plant Science. 2015,20(11), 741-753.).
- 5)It will be better to mention the gene feature of S5, Sa, HSA1, S7, DPL1/DPL2 and S27/S28 which helps readers to understand the progress in this area.

Reviewer #2 (Remarks to the Author):

The manuscript by Shen et al is an interesting and competent exploration of the hybrid male sterility causing by the gene at Sc locus. The results are well supported, elegant displayed in nice figures, and the conclusions are robust and of broad interest. Below is my only technical concern. Overall, this is a very good contribution to crop male sterility that will not only constitute relevant reading for the plant scientists but possess the vast utility potential in crop yield increase through hybrid vigor.

We can notice obviously that the chromatin fiber length of the 28kb Sc-i in BAC and genomic DNA are different in fiber-FISH. For example, in Fig 1d, the Sc-ib1/ib2 copies in MH63 BAC are about 4-5 μm and are clearly more than 6 μm (~8 μm ?). Actually, similar results have been reported and were revealed causing by the different extension levels of BAC and genomic DNA fibers. It is understandable for the readers with the fiber-FISH experience but may cause confusion for others. So, more explanations here are necessary to make it understandable for all readers. A better way is to summarize the fiber lengths of all the Sc-ib1/ib2 copies in BAC and genomic DNA, and we will see, as expected, the consistent length in each circumstance.

A few corrections or comments:

Line 282, “The BAC DNA was diluted to the appropriate concentration in AF solution”----- what is the meaning of “appropriate concentration” and what is that?

Line 17, “Hybrid sterility”----- “Hybrid sterility (HS) ”, a citation “HS” was found in ABSTRACT.

Line 113, “..... DNA fiber-FISH (fluorescence in situ hybridization).....”-----“DNA fiber-fluorescence in situ hybridization (fiber-FISH)”

Lines 287, 291, 296, “Probe 1----- Probe I”; “Probe 2----- Probe II”

Line 287, “Carnoy’s solution (ethanol: glacial acetic acid, 3:1)”----- “Carnoy’s solution (ethanol: glacial acetic acid, 3:1,v/v)”

Lines 290, 301, 310, 312 and 314, The companies and locations of the chemicals and software should be added to keep consistent formation trough the text.

Reviewer #3 (Remarks to the Author):

The authors of the manuscript entitled "Genomic structural variation-mediated allelic suppression causes hybrid male sterility in rice" showed that (1) Sc locus for hybrid pollen sterility in japonica and indica rice hybrid encodes a novel DUF1618-domain protein essential for pollen development, (2) duplicated genes, Sc-ib1, Sc-ib2, Sc-ia, acquired novel regulatory/promoter sequences and they suppress expression of the DUF1618-domain genes regulated under Sc-j promoter in F1 or heterozygotes at Sc (allelic suppression), and (3) allelic transcriptional suppression induces gametophytic pollen sterility in dosage-effect manners of Sc-ib1, Sc-ib2, and Sc-ia caused by copy number variations of ~28-kb segmental duplications, and (4) novel neutral alleles of Sc is possible to be developed by reducing dosage ratio of Sc-ib1, Sc-ib2, and Sc-ia by CRISPR/Cas9 knock-out.

Of the four main claims, the most important novelty of this manuscript is dosage-effect of allelic suppression for hybrid incompatibility, which is expected to be widely attractive from the reader of "Nature Communication". The data quality to reach to the claim seems satisfied for publication.

I found several things that minor correction is necessary.

Line 221-226

The "Killer-Protector" model is one of the way to interpret genetic mechanism of hybrid sterility at gene complex.

Why the Sc system should be forcefully explained under conceptual framework of "killer" and "protector" ?

As the authors might mention it (p 211~217), it may be suggested that expression level of canonical DUF1618-domain genes (Sc-j) is regulated by negative feedback of DUF1618 protein amount directly or DUF1618-related pathway and novel Sc-ib1/2 and Sc-ia promoters is incompatible to negative-feedback system. In this case, the word "protector" is not always appropriate.

The reviewer did not feel necessity of analogical explanation of Sc to the "Killer-Protector" model.

"killer" and "protector" are just words.

Line 223

In *Oryza rufipogon* gene pool, Sc-j and Sc-i types have differentiated.

Which of Sc-j and Sc-i is the ancestral type in AA genome species ?

By a parsimony principle, Sc-j seems ancestral type for me. But there is no evidence.

Depending on this, interpret of the BDM model and evolutionary pathway of reproductive isolation by Sc may change.

Investigation of Sc-j-like and Sc-i-like alleles in some other AA genome wild species would reveal ancestral types.

Line 223

In addition, reason why Sc system is "typical" Bateson-Dobzhansky-Muller type hybrid sterility is not sufficient.

For other reader, the authors should clearly explain what is BDM model and how the Sc system can be

corresponded to the BDM model.

In the one-locus BDM model, changing of allelic states $A \Rightarrow A' \Rightarrow A'$ should be explained to say novel mutation is neutral from natural selection. Between $A \Leftrightarrow A'$, and between $A' \Leftrightarrow A'$ hybrid sterility is not exhibited.

In this study, the reviewer understood that A is Sc-j, A' is DUF1618 genes acquiring the novel promoter (before gene duplications), and A' is Sc-i alleles having two or three duplication of DUF1618 with the novel promoter.

Personally, the reviewer considers that this is not BDM model in strict sense and this type of evolutionary step should be classified into the model different from BDM.

If the authors have different way of interpret between Sc and BDM models, please discuss more in details.

That' all. This paper is excellent work.

Reviewer #4 (Remarks to the Author):

The manuscript by Rongxin Shen et al. reports the genetic structure underlying variation at the Sc locus between rice subspecies indica and japonica. In crosses between these subspecies, the Sc locus contributes to hybrid sterility. The authors find that alleles at Sc in indica and japonica are highly divergent, with a single copy being present in japonica but multiple copies in indica. Through a series of detailed experiments, they infer that expression differences at Sc between indica and japonica lead to allelic suppression of the japonica allele in the F1. They further show through crossing experiments and CRISPR/Cas9 genome editing that variation in copy number at the Sc locus from indica contributes to the strength of hybrid sterility. These results are highly interesting and provide novel insights into the genetic basis of HS in plants. Unfortunately, it remains open how copy-number variation relates to allelic suppression and what the molecular

mechanism underlying allelic suppression is. The model proposed, however, provides a testable hypothesis.

The authors claim that in "anthers of the hybrids, high-expression of the Sc-i copies causes suppression of Sc-j expression". From the data and discussion presented, it is not clear whether this suppression really happens in the anthers. It seems that suppression (or rather the addition of (presumably) epigenetic marks that may ultimately cause suppression) could already happen at an earlier stage. Here I suggest that wording in the lead abstract be adjusted.

More importantly, however, I lack a discussion of how findings at the Sc locus relate to results obtained on other HS loci that have been characterized in rice. Also, from an evolutionary perspective it is highly interesting to see that *O. rufipogon* is polymorphic for the divergent *indica* and *japonica* alleles. This suggests the existence of polymorphic incompatibility loci in *O. rufipogon* - a topic that is of substantial interest to evolutionary biologists. Even more interesting may be the question how such divergent alleles can be maintained in *O. rufipogon*?

To warrant publication in a high impact journal, a study should ideally influence thinking in the field. From my perspective, the proposed model marks a step into this direction. Beyond this aspect, however, the manuscript seems to present a pretty closed story, even though it raises many highly interesting new questions - as any good study should. I would greatly appreciate it if the authors could improve further on this aspect.

Below I make a couple of specific comments that may help the authors to further improve their manuscript.

Introduction

It is not clear from the background provided why the mentioned strong heterosis in crosses between *indica* and *japonica* subspecies is prevented by hybrid sterility (HS) and can therefore not be utilized. Consequently, the first sentence in the lead paragraph is confusing to people not familiar with this particular system. In general, heterosis is strongest in the F1 generation. HS in crosses between *indica* and *japonica* should only matter then if pollen sterility in the F1 prevents seed formation and thus reduces yield. Is this indeed the problem? With respect to the issue of hybrid breeding, I agree that HS causes a major problem.

Lines 51-52: Why do molecular mechanisms for HS still remain largely obscure even though multiple loci underlying HS have been cloned? Please explain briefly.

Lines 59-60: In my view, CNVs are not an additional type of structural variations, such as insertions, deletions, etc. but instead are caused by insertions, deletions, etc.

Lines 62-64: Yes, the roles of CNVs in reproductive isolation require substantially more study but there are a few nice case studies in other systems published that could be cited here.

Line 65: Please add a few words describing the Sc locus, including e.g. the phenotype associated with this locus.

Results

Line 85: what is the reason for providing a link to the NCBI website?

Line 124: The authors state that " Sc-j in T65 was expressed specifically in anthers". I fail to see a signal (band) in Supplementary Fig. 7a.

Discussion

It is not clear from the present study whether Sc-i and Sc-j type alleles have indeed "evolved into a speciation gene" in japonica and indica subspecies. Apparently, *O. rufipogon* is polymorphic for these alleles and it seems that the two allele types were fixed in indica and japonica, respectively, during domestication. Crosses between these subspecies now lead to HMS. The question that should be asked is whether crosses between *O. rufipogon* accessions that are of Sc-i and Sc-j type, respectively, mirror HMS observed in crosses between japonica and indica subspecies? If yes, then no new evolution into a speciation gene has occurred in japonica and indica subspecies. Instead, *O. rufipogon* is polymorphic for incompatible alleles at the Sc locus. Intraspecific polymorphism for incompatibility alleles is a highly relevant topic and could be discussed in this study.

Methods

Line 252: Please indicate the subspecies to which these lines (e.g. MH63) belong

Line 309: Please specify the tissue from which RNA was extracted for expression analysis of the Sc locus, as well as the rice line(s) (or crosses) used.

Lines 320 ff - Quantification of the ratio of Sc-j over Sc-i copy numbers and transcript levels in the F1. Please refer to Supplementary Figure 8 here (ie. on line 323).

Data availability: The nucleotide sequence of Sc-j of T65 is essential to follow this study and also needs to be deposited and made publicly available.

Reviewers' comments:

Reviewer #1

Hybrid sterility is frequently observed in hybrids of japonica and indica rice subspecies. Shen et al., reported that the genomic structural change and copy number variation of *Sc* locus are responsible for japonica-indica hybrid male sterility. The japonica allele *Sc-j* encodes a DUF1618-domain protein which is required for normal pollen development, whereas the counterpart of indica *Sc-i* allele has two or three tandem-duplicated ~28-kb segments, and each containing a *Sc-j*-homolog but with a varied and recombinant promoter. The author observed that high-expression of the *Sc-i* copies in hybrid anthers result in the suppressed expression of *Sc-j*, causing aborted pollen development. The authors also validated this conclusion by that the transgenic plants of knock-outting two of the three *Sc-i* copies using CRISPR/Cas9 editing have the recovered expression of *Sc-j* and male fertility. This work provides insight into hybrid incompatibility in plants, will be useful for overcoming the reproductive barrier for crop breeding.

- 1) The authors investigate the expression of *Sc* using qRT-PCR in anthers. I suggest to check the expression in other tissues and in situ analysis is suggested for testing which cell type *Sc* expressed. The authors claimed that the *Sc* gene is a gametophytic factor for pollen development, it is essential to know the detailed cell type of the gene expression.

Response: Thank you for the suggestion. In fact we did in situ analysis of *Sc-j* expression in the early-bicellular pollen (the expression peak stage) of japonica rice T65, but failed to detect clear signal, probably because of the very low level of the *Sc-j* transcript (only 2% of the *OsActin 1* transcript level, see Figure 2c). At the early-bicellular pollen stage, the tapetum of anthers is completely degenerated and disappeared; thus, the peak expression of *Sc-j* detected in anthers of early-bicellular pollen stage may be attributed from the pollen grains (note that the relative value in pollen grains was about 2 folds of that in the whole anthers that contain pollen and anther cell wall, see the revised Figure 3a). We redone the experiment to separate the pollen grains and anther wall cells (epidermis) of T65 and did qRT-PCR, and the results indicate that *Sc-j* was expressed mainly in the isolated pollen grains but less in anther wall cells, which is shown in the revised Figure 2b.

- 2) The authors claimed that *Sc* is required for pollen formation. But the evidence on the role of *Sc* in pollen development is not enough. I suggest the authors to check the anther developmental changes in the male fertile and sterile lines by sectioning anthers and pollen grains.

Response: We show that suppression of *Sc-j* expression in anthers (in half of pollen grains) of the transgenic plants with hemizygous antisense-transgene produced sterility of half of pollen grains (semi-sterility) (Figure 2e, f), and the antisense-transgene in the progenies of the lines segregated as 1: 1 (presence : absence) (Supplementary Figure 9). These results indicate that *Sc-j* is required for pollen formation, and acts as gametophytic factor. The observation of developmental changes of the *Sc-j*-carrying pollen grains in sectioning anthers of the hybrid (T65/E5 cross) has been reported by Dr. Xiangdong Liu (South China Agricultural University) (cited in the reference 28 of the original manuscript); the

***Sc-j*-carrying pollen grains become abnormally developed at the bicellular stage. In this revised manuscript, we describe the abnormality of the sectioning pollen in more detail according to this report.**

3) To confirm the function of *Sc* in pollen development, is it possible to use genome editing method instead of RNAi approach?

Response: The antisense and RNAi methods also are effective and widely used for study of gene functions. We carried out the experiment of antisense-based suppression of *Sc-j* in many years ago (2008-2010), and obtained the results that show clearly the phenotype (sterility in ~50% pollen grains in the hemizygous transgenic plants) and 1:1 segregation of the antisense-transgene in the progenies that suggests the pollen grains carrying the antisense-transgene were sterile and indicates the gametophytic feature of the gene. Anyway, we are carrying out CRISPR/Cas9 editing of this gene to further study its mechanism in regulation of pollen development.

4) To understand the mechanism on how the hybrid is male sterile, is it possible to checked the expression of the genes known to be required rice male development (reviewed by Zhang and Liang: Improving food security: Using male fertility for hybrid seed breeding. Science. 2016, sponsored collection: 45; Shi et al., Trends in Plant Science. 2015,20 (11), 741-753.).

Response: Thank you for this suggestion. We analyzed the expression of 14 genes known for microspore/pollen development in anthers of the hybrid and parent T65. The results are presented in Supplementary Figure 10. The reference (Shi et al., 2015) also was cited in the revised manuscript.

5) It will be better to mention the gene feature of *S5*, *Sa*, *HSA1*, *S7*, *DPL1/DPL2* and *S27/S28* which helps readers to understand the progress in this area.

Response: Thank you for the suggestion. We described the working models of four of these gene loci in the revised manuscript in lines 53-78.

Reviewer #2

The manuscript by Shen et al is an interesting and competent exploration of the hybrid male sterility causing by the gene at *Sc* locus. The results are well supported, elegant displayed in nice figures, and the conclusions are robust and of broad interest. Below is my only technical concern. Overall, this is a very good contribution to crop male sterility that will not only constitute relevant reading for the plant scientists but possess the vast utility potential in crop yield increase through hybrid vigor.

We can notice obviously that the chromatin fiber length of the 28kb *Sc-i* in BAC and genomic DNA are different in fiber-FISH. For example, in Fig 1d, the *Sc-ib1/ib2* copies in MH63 BAC are about 4-5 μm and are clearly more than 6 μm (~8 μm ?). Actually, similar results have been reported and were revealed causing by the different extension levels of BAC and genomic DNA fibers. It is understandable for the readers with the fiber-FISH experience but may cause confusion

for others. So, more explanations here are necessary to make it understandable for all readers. A better way is to summarize the fiber lengths of all the Sc-ib1/ib2 copies in BAC and genomic DNA, and we will see, as expected, the consistent length in each circumstance.

Response: Thank you for the suggestion. We explained this difference for fiber-FISH of BAC and gDNA in the figure legends of Figure 1 in the revised manuscript.

A few corrections or comments:

Line 282, "The BAC DNA was diluted to the appropriate concentration in AF solution"-----what is the meaning of "appropriate concentration" and what is that?

Line 17, "Hybrid sterility" ----- "Hybrid sterility (HS)", a citation "HS" was found in ABSTRACT.

Line 113, "..... DNA fiber-FISH (fluorescence in situ hybridization)....." ----- "DNA fiber-fluorescence in situ hybridization (fiber-FISH)"

Lines 287, 291, 296, "Probe 1----- Probe I" ; "Probe 2----- Probe II"

Line 287, "Carnoy' s solution (ethanol: glacial acetic acid, 3:1)" ----- "Carnoy' s solution (ethanol: glacial acetic acid, 3:1,v/v)"

Lines 290, 301, 310, 312 and 314, The companies and locations of the chemicals and software should be added to keep consistent formation through the text.

Response: We corrected these words.

Reviewer #3

The authors of the manuscript entitled "Genomic structural variation-mediated allelic suppression causes hybrid male sterility in rice" showed that (1) Sc locus for hybrid pollen sterility in japonica and indica rice hybrid encodes a novel DUF1618-domain protein essential for pollen development, (2) duplicated genes, Sc-ib1,Sc-ib2,Sc-ia, acquired novel regulatory/promoter sequences and they suppress expression of the DUF1618-domain genes regulated under Sc-j promoter in F1 or heterozygotes at Sc (allelic suppression), and (3) allelic transcriptional suppression induces gametophytic pollen sterility in dosage-effect manners of Sc-ib1,Sc-ib2, and Sc-ia caused by copy number variations of ~28-kb segmental duplications, and (4) novel neutral alleles of Sc is possible to be developed by reducing dosage ratio of Sc-ib1,Sc-ib2, and Sc-ia by CRISPR/Cas9 knock-out.

Of the four main claims, the most important novelty of this manuscript is dosage-effect of allelic suppression for hybrid incompatibility, which is expected to be widely attractive from the reader of "Nature Communication". The data quality to reach to the claim seems satisfied for publication.

I found several things that minor correction is necessary.

Line 221-226

The "Killer-Protector" model is one of the way to interpret genetic mechanism of hybrid sterility at gene complex. Why the Sc system should be forcefully explained under conceptual framework of "killer" and "protector" ?

As the authors might mention it (p 211~217), it may be suggested that expression level of

canonical DUF1618-domain genes (*Sc-j*) is regulated by negative feedback of DUF1618 protein amount directly or DUF1618-related pathway and novel *Sc-ib1/2* and *Sc-ia* promoters is incompatible to negative-feedback system. In this case, the word "protector" is not always appropriate. The reviewer did not feel necessity of analogical explanation of *Sc* to the "Killer-Protector" model. "killer" and "protector" are just words.

Response: Thank you for the comments. The establishment of the best fitted model for *Sc* needs further study, so at present we do not link *Sc* to the Killer-protector model in the revised manuscript.

Line 223

In *Oryza rufipogon* gene pool, *Sc-j* and *Sc-i* types have differentiated. Which of *Sc-j* and *Sc-i* is the ancestral type in AA genome species? By a parsimony principle, *Sc-j* seems ancestral type for me. But there is no evidence. Depending on this, interpretation of the BDM model and evolutionary pathway of reproductive isolation by *Sc* may change. Investigation of *Sc-j*-like and *Sc-i*-like alleles in some other AA genome wild species would reveal ancestral types.

Response: From the sequence compositions of the promoters of these gene copies (*Sc-j* and *Sc-is* have the same promoter sequence and *Sc-ia/ib* have the promoter sequence that might have resulted from recombination), *Sc-j* and *Sc-is* are really allelic thus they seem ancestral type, and *Sc-ia/ib* may be the lately diverged type. However, we need to extensively investigate the sequence variation (by PCR/sequencing) and CNV (by PCR/fiber-FISH) of the *Sc* locus in accessions of the wild and cultivated *Oryza* species, to understand its evolution.

Line 223

In addition, reason why *Sc* system is "typical" Bateson-Dobzhansky-Muller type hybrid sterility is not sufficient. For other reader, the authors should clearly explain what is BDM model and how the *Sc* system can be corresponded to the BDM model.

In the one-locus BDM model, changing of allelic states $A \Rightarrow A' \Rightarrow A''$ should be explained to say novel mutation is neutral from natural selection. Between $A \Leftrightarrow A'$, and between $A' \Leftrightarrow A''$ hybrid sterility is not exhibited. In this study, the reviewer understood that *A* is *Sc-j*, *A'* is DUF1618 genes acquiring the novel promoter (before gene duplications), and *A''* is *Sc-i* alleles having two or three duplication of DUF1618 with the novel promoter.

Personally, the reviewer considers that this is not BDM model in strict sense and this type of evolutionary step should be classified into the model different from BDM.

If the authors have different way of interpretation between *Sc* and BDM models, please discuss more in details.

That's all. This paper is excellent work.

Response: Thank you for the evaluation and comments. As mentioned above, we need to extensively investigate the sequence variation of the *Sc* locus in the *Oryza* genus to study the evolution of *Sc*, in this paper we do not define the evolutionary model of this locus as a BDM type, but *Sc* can be used to study a BDM or novel model of single-locus hybrid incompatibility (lines 276-283)..

Reviewer #4

The manuscript by Rongxin Shen et al. reports the genetic structure underlying variation at the *Sc* locus between rice subspecies *indica* and *japonica*. In crosses between these subspecies, the *Sc* locus contributes to hybrid sterility. The authors find that alleles at *Sc* in *indica* and *japonica* are highly divergent, with a single copy being present in *japonica* but multiple copies in *indica*. Through a series of detailed experiments, they infer that expression differences at *Sc* between *indica* and *japonica* lead to allelic suppression of the *japonica* allele in the F1. They further show through crossing experiments and CRISPR/Cas9 genome editing that variation in copy number at the *Sc* locus from *indica* contributes to the strength of hybrid sterility. These results are highly interesting and provide novel insights into the genetic basis of HS in plants. Unfortunately, it remains open how copy-number variation relates to allelic suppression and what the molecular mechanism underlying allelic suppression is. The model proposed, however, provides a testable hypothesis.

The authors claim that in "anthers of the hybrids, high-expression of the *Sc-i* copies causes suppression of *Sc-j* expression". From the data and discussion presented, it is not clear whether this suppression really happens in the anthers. It seems that suppression (or rather the addition of (presumably) epigenetic marks that may ultimately cause suppression) could already happen at an earlier stage. Here I suggest that wording in the lead abstract be adjusted.

More importantly, however, I lack a discussion of how findings at the *Sc* locus relate to results obtained on other HS loci that have been characterized in rice. Also, from an evolutionary perspective it is highly interesting to see that *O. rufipogon* is polymorphic for the divergent *indica* and *japonica* alleles. This suggests the existence of polymorphic incompatibility loci in *O. rufipogon* - a topic that is of substantial interest to evolutionary biologists. Even more interesting may be the question how such divergent alleles can be maintained in *O. rufipogon*?

To warrant publication in a high impact journal, a study should ideally influence thinking in the field. From my perspective, the proposed model marks a step into this direction. Beyond this aspect, however, the manuscript seems to present a pretty closed story, even though it raises many highly interesting new questions - as any good study should. I would greatly appreciate it if the authors could improve further on this aspect.

Response: Thank you for the evaluation and comments. With your comments that the effective interaction of *Sc-i* with *Sc-j* may not firstly occur at the anthers but also possibly at an earlier stage, although *Sc-i* is expressed with much higher level at anthers than in young panicles (see Supplementary Figure 7a). We revised the abstract and text not to claim the possible interaction in anthers. In the revision we discussed the possible polymorphic for the divergent *Sc-j-like* and *Sc-i-like* alleles in wild and cultivated rice populations and their genetic effects. We also discussed the diverse and common molecular genetic mechanisms of HS loci in plants.

Below I make a couple of specific comments that may help the authors to further improve their manuscript.

Introduction

It is not clear from the background provided why the mentioned strong heterosis in crosses between *indica* and *japonica* subspecies is prevented by hybrid sterility (HS) and can therefore not be utilized. Consequently, the first sentence in the lead paragraph is confusing to people not familiar with this particular system. In general, heterosis is strongest in the F1 generation. HS in

crosses between indica and japonica should only matter then if pollen sterility in the F1 prevents seed formation and thus reduces yield. Is this indeed the problem? With respect to the issue of hybrid breeding, I agree that HS causes a major problem.

Response: *Japonica-indica* hybrids have strong hybrid vigor in vegetative organs, but the hybrids also produce severe HS (male and female sterility) due to the accumulated effects of multiple HS loci, which prevents seed setting and thus reduces grain yield. We improve the Abstract/Introduction to better describe the relation between HS and utilization of heterosis for increasing grain yield.

Lines 51-52: Why do molecular mechanisms for HS still remain largely obscure even though multiple loci underlying HS have been cloned? Please explain briefly.

Response: In *japonica* and *indica* rice cultivars, about 50 loci for HS have been reported, while at present only the genes at five loci (including a pair of loci) for the *japonica-indica* HS were cloned, and the molecular genetic mechanisms of some of them remain not to be evaluated. We revised the description for this.

Lines 59-60: In my view, CNVs are not an additional type of structural variations, such as insertions, deletions, etc. but instead are caused by insertions, deletions, etc.

Response: In many CNV cases, a relatively large genomic fragment containing one or more complete genes are tandemly duplicated for two or more copies, which can produce a direct effect of increased gene dosage (thus higher expression levels) and causes phenotypic variation. Thus, in the literature, CNVs are widely defined as a special type of genomic structural variation, which has different feature to those of the insertion or deletion of a single copy of genomic fragment. In the case of *Sc-i*, we show that the different degrees of *Sc-i* CNV (e.g., two and three copies) caused variation on the hybrid male sterility (different degrees of transmission ratio distortion).

Lines 62-64: Yes, the roles of CNVs in reproductive isolation require substantially more study but there are a few nice case studies in other systems published that could be cited here.

Response: We removed the citations describing CNVs on phenotypic variations from Discussion to this paragraph.

Line 65: Please add a few words describing the *Sc* locus, including e.g. the phenotype associated with this locus.

Response: revised accordingly (lines 93-96).

Results

Line 85: what is the reason for providing a link to the NCBI website?

Response: This sentence was revised as "Analysis of the genome sequence of an *indica* cultivar 93-11 in GeneBank.

Line 124: The authors state that " *Sc-j* in T65 was expressed specifically in anthers". I fail to see a signal (band) in Supplementary Fig. 7a.

Response: The blight of this photograph was adjusted to able to see.

Discussion

It is not clear from the present study whether Sc-i and Sc-j type alleles have indeed "evolved into a speciation gene" in japonica and indica subspecies. Apparently, *O. rufipogon* is polymorphic for these alleles and it seems that the two allele types were fixed in indica and japonica, respectively, during domestication. Crosses between these subspecies now lead to HMS. The question that should be asked is whether crosses between *O. rufipogon* accessions that are of Sc-i and Sc-j type, respectively, mirror HMS observed in crosses between japonica and indica subspecies? If yes, then no new evolution into a speciation gene has occurred in japonica and indica subspecies. Instead, *O. rufipogon* is polymorphic for incompatible alleles at the Sc locus. Intraspecific polymorphism for incompatibility alleles is a highly relevant topic and could be discussed in this study.

Response: We added a sentence "It is interesting to know that if there are other intermediate Sc-i-like and/or Sc-j-like, hybrid-neutral or -incompatible alleles in the wild and cultivated rice populations, for example the primary diverged DUF1618 gene that acquired the novel promoter but without the segmental duplication" in the Discussion (line2 276-280). In fact, we are doing extensively investigation of the sequence variation and CNV of the Sc locus in the *Oryza* genus (the wild and cultivated species), to understand its evolution (however, determination of the segmental copy number by genomic Fiber-FISH is low-efficient and very time-consuming). In addition, we are doing a cross between an *O. rufipogon* accession containing a Sc-i-like allele and a japonica line; however, we need 2.5 years to obtain F₂ plants for analysis of the segregation ratio (one year for crossing, one year for F₁ planting to get F₂ seeds, due to the strong photoperiod sensitivity for heading). These results, if valuable, will be published in the future.

Methods

Line 252: Please indicate the subspecies to which these lines (e.g. MH63) belong.

Line 309: Please specify the tissue from which RNA was extracted for expression analysis of the Sc locus, as well as the rice line(s) (or crosses) used.

Lines 320 ff - Quantification of the ratio of Sc-j over Sc-i copy numbers and transcript levels in the F₁. Please refer to Supplementary Figure 8 here (ie. on line 323).

Data availability: The nucleotide sequence of Sc-j of T65 is essential to follow this study and also needs to be deposited and made publicly available.

Response: We revised these, and provided sequence of Sc-j of T65 to GenBank.

Reviewers' comments:

Reviewer #1 (Remarks to the Author):

I appreciate the efforts from the authors in revising this manuscript to address my comments. In this revised version I only have one concern about the in situ data. For the in situ test, the authors claimed the no detectable signal at stage 11 during the early-bicellular stage. In Figure 2 the expression of Sc gene seems to be higher from stage 7 to 10 than that of stage 11 using qRT-PCR in anthers. Did the authors try to use the samples of early stages for in situ analysis? Even though the authors claimed that Sc is a gametophytic factor. But the data on expression tissue of Sc is quite important for functional understanding of Sc. Further, the detectable defect of the hybrid may be caused by the internal signal occurred at the early stage.

Reviewer #2 (Remarks to the Author):

All my comments were addressed and no further concerns for me now.

Reviewer #3 (Remarks to the Author):

I confirmed the authors correctly responded to the reviewer's comment in the manuscript.

Authors' response to the "Line 221-226 in the original submission".

I confirmed the authors' case does not link to the "Killer-Protector" model.

This correction probably made the reader more directly understand phenomena in this manuscript without any confusion with Killer-Protector mode, although possibility still remains that other genetic factors adjacent to Sc are associated to.

Authors' response to the "Line 223 in the original submission".

I can understand authors' current idea for the evolutionary way of this locus at Line 276-282 in the revised manuscript. I hope evolutionary steps of Sc locus will be elucidated.

Including the above, I noticed several points that should be corrected.

L114

GeneBank may be a misspelling for GenBank.

L266

"Gamete elimination" have a special meaning in rice hybrid sterility like S1.
The sterility allele (gamete eliminator) affect male and female (both sex) gametes
to induce genotype-specific elimination of gametes.
"male-gamete elimination" is possibly better.

Reviewer #4 (Remarks to the Author):

From my perspective, responses to reviewer comments and requests are largely appropriate, except the response concerning line 85 in results and the new sentence added in response to my comment on the discussion (see also last comment below).

However, I have a number of suggestions on how wording could be revised:

Line 35: 'proposes' instead of 'proposed'

Line 36: In the classical two-locus model, the two loci are not required to be related. I thus propose removing 'of related loci'.

Line 40: Replace 'thus' by 'and'.

Line 43: 'has contributed' instead of 'have contributed'.

Line 77 and 78: Suggested rewording: " Even though much has been learned about the genetic basis of HS in rice, we still lack a complete picture of its complexity and methods to overcome HS for future hybrid rice breeding"

Line 82: Replace 'have made only less' by 'have had only limited'

Lines 93 and following - suggested rewording: Sc is a major locus conferring HMS in japonica-indica hybrids and was identified using a cross between the japonica cultivar Taichung 65 (T65, carrying a japonica allele Sc-j) and its near-isogenic line E5 (which has the T65 nuclear background but contains an indica allele Sc-i that was introgressed from the indica cultivar Peh-ku).

Lines 106-107: Suggestion: So far, no neutral allele at the Sc locus have been identified in rice germplasm.

Line 114: It is unclear what 'GeneBank' means here. The information provided is insufficient. In case authors refer to NCBI, then this is written as 'GenBank' and appropriate accession numbers should be provided.

Line 273: The wording ' the Sc-i allele variants have the role to specifically kill' implies a purpose, which is certainly not given. Please reword.

Lines 276-280: This sentence appears incomplete. Please revise.

Responses to reviewers' comments:

Reviewer #1:

I appreciate the efforts from the authors in revising this manuscript to address my comments. In this revised version I only have one concern about the *in situ* data. For the *in situ* test, the authors claimed the no detectable signal at stage 11 during the early-bicellular stage. In Figure 2 the expression of *Sc* gene seems to be higher from stage 7 to 10 than that of stage 11 using qRT-PCR in anthers. Did the authors try to use the samples of early stages for *in situ* analysis? Even though the authors claimed that *Sc* is a gametophytic factor. But the data on expression tissue of *Sc* is quite important for functional understanding of *Sc*. Further, the detectable defect of the hybrid may be caused by the internal signal occurred at the early stage.

Response: Thank you very much for the comments. We agree with the reviewer that the data on the expression pattern of *Sc* is important for functional understanding of the gene. We had made efforts to address the expression patterns of the *Sc* alleles and apologize for the ambiguity of our statement on this issue.

At the *Sc* locus, the *japonica* allele gene *Sc-j* and the recombinant type of the *indica* allele *Sc-i* have totally different expression patterns and levels. Our qRT-PCR results (Fig. 2a) show that *Sc-j* is specifically expressed in the early-bicellular anthers (stage 11) at a low level (and silence or more lower levels before and after the early-bicellular stages), which is consistent with the microarray transcriptome data for *Sc-j*; in this revised manuscript we added the microarray profile of *OsActin 1* as a reference to give the relative expression level (0.041) of *Sc-j* to *OsActin 1* (see Supplementary Fig. 7b), which is consistent with our qRT-PCR results (Fig. 2b and Fig. 3a). By contrast, *Sc-i* is broadly expressed in various tissues and at high levels (Fig. 2a and Supplementary Figure 7a), probably due to its different promoter sequences and repeated copies. To understand the basic function of the *Sc* gene, it will help to know the *in situ* expression of the putatively primitive type *Sc-j*, rather than the varied type *Sc-i* that has the altered constitutive/high-level expression pattern. However, due to the low-level expression of *Sc-j*, we failed to successfully obtain clear signals of *in situ* mRNA hybridization in anthers of various stages of T65, although we tried this experiment many times. As an alternative method, we then used qRT-PCR to analyze mRNAs from isolated early-bicellular pollen grains and the anther wall cells of the same stage. Our new data confirmed the specific expression of *Sc-j* in early-bicellular pollen grains, but not in anther wall cells (Fig. 2b). Finally, our *Sc-j*-antisense transgenic plants and their segregation analysis (e.g., 1:1 ratio of the presence: absence of the antisense-transgene in the T₁ families) further show the gametophytic function of *Sc-j* (Fig. 3c, d and Supplementary Figure 9).

To avoid the lack clarity on this point, we revised the text properly. We believe that these results and improvements will more clearly describe the pollen-specific/low-level expression and the gametophytic function of *Sc-j*.

Reviewer #2:

All my comments were addressed and no further concerns for me now.

Response: We thank you for your time.

Reviewer #3:

I confirmed the authors correctly responded to the reviewer's comment in the manuscript.

Authors' response to the "Line 221-226 in the original submission". I confirmed the authors' case does not linked to the "Killer-Protector" model. This correction probably made the reader more directly understand phenomena in this manuscript without any confusion with Killer-Protector mode, although possibility still remain that other genetic factors adjacent to *Sc* is associated to Authors' response to the "Line 223 in the original submission". I can understand authors' current idea for the evolutionary way of this locus at Line 276-282 in the revised manuscript. I hope evolutionary steps of *Sc* locus will be elucidated. Including the above, I noticed several points that should be corrected.

L114

GeneBank may be a misspelling for GenBank.

L266

"Gamete elimination" have a special meaning in rice hybrid sterility like *S1*. The sterility allele (gamete eliminator) affect male and female (both sex) gametes to induce genotype-specific elimination of gametes. "male-gamete elimination" is possibly better.

Response: Thank you for the comments. We have revised the manuscript accordingly.

Reviewer #4:

From my perspective, responses to reviewer comments and requests are largely appropriate, except the response concerning line 85 in results and the new sentence added in response to my comment on the discussion (see also last comment below).

However, I have a number of suggestions on how wording could be revised:

Line 35: 'proposes' instead of 'proposed'.

Line 36: In the classical two-locus model, the two loci are not required to be related. I thus propose removing 'of related loci'.

Line 40: Replace 'thus' by 'and'.

Line 43: 'has contributed' instead of 'have contributed'.

Line 77 and 78: Suggested rewording: " Even though much has been learned about the genetic basis of HS in rice, we still lack a complete picture of its complexity and methods to overcome HS for future hybrid rice breeding"

Line 82: Replace 'have made only less' by 'have had only limited'

Lines 93 and following - suggested rewording: *Sc* is a major locus conferring HMS in japonica-indica hybrids and was identified using a cross between the japonica cultivar Taichung 65 (T65, carrying a japonica allele *Sc-j*) and its near-isogenic line E5 (which has the T65 nuclear background but contains an indica allele *Sc-i* that was introgressed from the indica cultivar Peh-ku).

Lines 106-107: Suggestion: So far, no neutral allele at the *Sc* locus have been identified in rice germplasm.

Line 114: It is unclear what 'GeneBank' means here. The information provided is insufficient. In case authors refer to NCBI, then this is written as 'GenBank' and appropriate accession numbers should be provided.

Line 273: The wording ' the *Sc-i* allele variants have the role to specifically kill' implies a purpose,

which is certainly not given. Please reword.

Lines 276-280: This sentence appears incomplete. Please revise.

Response: Thank you very much for these corrections and comments. We have revised the manuscript accordingly. We also changed “the *Sc-i* allele variants have the role to specifically kill pollen grains with *Sc-j* but not those carrying the *Sc-i* variants in the hybrids” to “and (the *Sc-i* allele variants) can cause specific abortion of pollen with *Sc-j* but not pollen carrying the *Sc-i* variants in the hybrids”. We added a sentence “The BDM-type genetic incompatibility results from the divergence of certain loci during evolution” (lines 285-286) before the description “It would be interesting to know if other intermediate types of *Sc-i*-like and/or *Sc-j*-like allele variants” (lines 286-290).